# Survival of Bacteriophage T4 in Quasi-Pure Ionic Solutions

**DOI:** 10.3390/v15081737

**Published:** 2023-08-14

**Authors:** Seiko Hara, Isao Koike

**Affiliations:** 1Miyazaki International College, 1405 Kano, Miyazaki 889-1605, Japan; 2Atmosphere and Ocean Research Institute, The University of Tokyo, Chiba 277-8564, Japan; isakoike@g.ecc.u-tokyo.ac.jp

**Keywords:** infectivity, neutral solution, acidic solution, alkaline solution, deprotonation

## Abstract

The preservative qualities of individual ionic compounds impacting the infectivity of T4 virions were elucidated. T4 virions were immersed in quasi-pure ionic solutions prior to the adsorption process, and the plaque forming unit (pfu) values of these were measured following the conventional method. In neutral ionic solutions, the minimum and the optimum concentrations of preservative qualities corresponded with the results obtained from the multi-ionic media/buffers. In acid and alkali solutions, phages show tolerances at a pH range of 5–11 in multi-ionic media/buffers. T4 virions show no tolerance in quasi-pure acid, neutral, and weak alkaline conditions. The preservative quality of T4 virions increased in over 10^−1^ mM OH^−^ solution, equivalent to a pH value over 10, which corresponds to the pKa of the deprotonation of the DNA bases G and T. Infectivity was lost below 10^−1^ mM OH^−^ and higher than 10 mM OH^−^. These results imply that maintaining infectivity of a virion may need the flexibility of the intra-capsid DNA by deprotonation.

## 1. Introduction

Understanding the responses of virions to ambient biological, chemical, and physical factors is critical not only for basic biological viewpoints, but also for defining the optimum and/or most harmful conditions for the industrial, medical, and agricultural use of virions [1,2]. Among these factors, chemism has been studied for a long time [3,4,5,6,7,8]. It is generally accepted that divalent cations, such as Ca^2+^ and Mg^2+^, affect the lytic cycle of many bacteriophages, acting on either adsorption, penetration, or intracellular development of the virus [2,6,9,10,11]. The adsorption on bacterial cells of some phages has also been accelerated by univalent cations, such as Na^+^ [4,8]. Phages can generally show infection capabilities at a pH range of 5–11 [1,12,13,14,15,16,17]. The reported results, however, were obtained in solutions of mixed ionic compounds, e.g., media and buffers. The operation of multi-ionic media and buffers on viral activity is the synthetic effect of the ionic components. Although elucidating the effects of solo-ionic compound solutions on viral activities is essential to understand the synthetic effects, the viral behaviors in solo-ionic conditions, e.g., the possibility of the survival of virions in pure acid or alkaline solutions, have been unknown. Accordingly, this study proposed to develop a new method to manipulate T4 virions in a solo-ionic condition and applied this method to elucidate the influences of solo-ions on infectivity.

The major findings of this study were as follows. Virions will be inactivated immediately in pure water [2,4,8]. In the new method, the amounts of ions derived from the solutes in the inoculant were controlled at the minimum level which did not affect the actions of the target solo-ion. This is the reason why we call the sole-ionic compound solutions quasi-pure. Based on this condition, the pfu spectra obtained from the quasi-pure ionic solutions generally agreed with previous reports in neutral conditions [2,6]. Contrary to the previous reports [1,12,13,14,15,16,17], T4 virions could survive, but only in 10^−1^–10 mM OH^−^ solution, equivalent to alkaline pH 10–12. No infectious activity was observed in acid to neutral and even in weak alkaline solutions. The activation-inactivation is switched on/off at 10^−4^ N OH^−^, pH 10 equivalent, which corresponds to the pKa of the deprotonation of the DNA bases G and T. The results imply that maintaining infectivity of a virion may depend on the flexibility of the intra-capsid DNA by deprotonation.

## 2. Materials and Methods

### 2.1. Strains

The bacteriophage studied was T4 (ATCC 11303-B4), and its host bacteria were *Escherichia coli* (ATCC 11303).

### 2.2. Preparation of T4 Virions

Peptone broth was used for culturing the host bacteria, *E. coli*. The T4 suspension was obtained by the plate lysate method and the small-scale liquid culture [18]. In the plate lysate method, the virions were suspended in 2–3 mL of an electrolyte solution including 1.8 mM NaCl, 0.12 mM MgSO_4_, 0.12 mM MgCl_2_, 0.034 mM CaCl_2_, and 0.05 mM KCl. The extracted suspension was sterilized by filtration of the eluent with 0.2 μm filter (Advantec AS020). To exclude the effects of anonymous ions, the suspensions of T4 virions were purified with ultracentrifugation and dialysis. For ultracentrifugation, crude bacteriophage particles were purified through isopycnic centrifugation through CsCl gradients [18], [Beckman XPN-90, SW32 rotor, 4 °C, 24 h, in 2019]. Following ultracentrifugation, T4 suspensions were dialyzed against T-buffer (modified TM buffer [18] 0.1 M NaCl, 2 mM MgSO_4_, 0.5 mM phosphate buffer, pH 7.5) for one week replacing the outside T-buffer five times. The chemicals used were special grade products from FUJIFILM Wako Pure Chemical Corp., Tokyo, Japan.

### 2.3. Preparation of Suspension of T4 Virions in Quasi-Pure Solution

It is well known that the infectious ability of virions is instantly lost when they are immersed in pure water [2,4,8], with the result that the active virions are practically stored in relevant ionic solutions. When the stored virions are used for experiments, aliquots of stored suspensions are inoculated in target solvents. Accordingly, the ions included in the aliquots of the suspensions are added into the target solutions. It is critical to maintain the concentrations of these ions introduced by the inoculations for studying the connections between the ionic concentrations and the viral activity. In these experiments, the ultra-centrifuged T4 virions were dialyzed against T-buffer and stored in T-buffer. After the stored virions were appropriately diluted with T-buffer, these T-buffer suspensions were diluted twice. First, they were diluted with Milli-Q water 100-fold and acclimated to this condition for five minutes. At this point, the ionic concentrations of the viral suspensions were 0.01 T-buffer. Second, the Milli-Q water dilutions were diluted again 100-fold with the test solutions. As a whole, the final concentrations of T-buffer in the test solutions were 0.0001 T-buffer, i.e., 0.01 mM NaCl, 0.2 μM MgSO_4_, and 0.05 μM phosphate buffer. In these diluted ionic conditions, the virions gradually lost their infectious ability by the time of their manipulation. The time course measurements of viral activities in these diluted conditions indicating the decrease of the activity were expressed as:*V_t_* = *V*_0_
*e^−kt^*(1)

The inactivation coefficient *–k* was defined by slope of the following linear Equation (2):*Ln*(*V_t_*) = *Ln*(*V_0_*) − *kt*(2)
where *V_t_*: density of active virions at *t*, *V*_0_: density of active virions at the initial moment, −*k*: inactivation coefficient, *t*: time. 

For counting plaque forming units, virions were plated after the acclimation at the first dilution for 5 min, followed by incubation in the second dilution for 15 min. 

### 2.4. Ionic Solutions

In this paper, we are concerned with the concentration of a solute. To prepare the acid and alkali solutions of the testing concentrations, 1 M solutions of acid and alkali were diluted to the target concentrations. According to the solubilities, the original solution of Ca(OH)_2_ was 10 mM, and it was 0.15 mM for Mg(OH)_2_. The pH value of the dilution was not measured each time, and pH was not adjusted. However, the pH values measured by Whatman pH-indicator paper were close to the expected values, e.g., the pH value of 1 mM NaOH was ca. pH 11. The pH values reported here, nominal pH, were not the measured values, but the calculated values from concentrations of H^+^ and OH^−^. On the other hand, the pH values of the solutions of neutral ionic compounds, i.e., NaCl, Na_2_SO_4_, KCl, K_2_SO_4_, CaCl_2_, CaSO_4_, MgCl_2_, and MgSO_4_, fell within the range of pH 6–6.5. These ionic compounds were selected as the combinations of the representative univalent and divalent cations and anions, as well as the major components of the cell electrolytes.

### 2.5. Plaque Forming Unit (pfu)

The plaque forming units (pfu) of each case were measured twice independently using two populations of T4, i.e., adjacent two fractions of ultra-centrifuged samples, which were expressed as thicker lines and open markers (the heavier fraction) and thinner lines and solid markers (the lighter fraction). Aliquots of virions were plated with *E. coli* on 1% agar peptone plates covered with 0.5% agar peptone top agar. Inoculated plates were incubated at 36 °C for 12 h before counting. The numbers of plaques were adjusted to 10~500 plaques per plate as was possible. The pfu values indicated in figures were % of the pfu values of T4 populations suspended in T-buffer. The pfu values of the stock T4 suspensions were enumerated frequently to confirm the activity of the T4 virion during the experiments. The statistical analysis of the parameters and model was obtained through the statistics package Data Analysis in Excel.

## 3. Results

### 3.1. Time Course of Viral Activity

Table 1 indicates the values of the inactivation coefficient (*−k*), the initial abundance of virins (*V*_0_), and R^2^ of Equation (1). As *p* values of *−k* in the Table 1 indicated, all the values of the inactivation coefficients (*−k*) were significant (*p* < 0.01) except for the case of 0.1 mM Mg(OH)_2_ solution. The infectivity of T4 virions in 0.1 mM Mg(OH)_2_ solution was stable over time; accordingly, the value of *−k* approximated 0, and the value of R^2^ became small. The *p* value of *−k* also indicated no significant correlation between the infectivity and time. At selected dilutions of T4 virions in T-buffer, in the first dilution in Milli-Q, when the ion concentration was 0.01 T-buffer, more than 95% of virions were active after 5 min. When the 0.01 T-buffer was diluted into Milli-Q again, the ionic composition was 0.0001 T-buffer. Virions in this condition lost the infectious ability instantly, which indicates that the ionic composition in 0.0001 T-buffer, i.e., 0.01 mM NaCl, 0.2 μM MgSO_4_, and 0.05 μM phosphate buffer, does not support the infectious ability of T4. The testing ionic compound was included in the solution of the second dilution, where even if the remaining T-buffer exists, the remnant of T-buffer does not disturb the infectious process in the testing ionic solution. Therefore, we call this condition “quasi-pure” and can practically measure the action of a sole additional compound on the infectious process. 

The incubation period of the test suspension was chosen as 15 min., because the survivals of virions in the test solutions of preferable ion concentrations were higher than 60% after 15 min. (see below), and the differences of the rates of survival between different ion concentrations became sufficiently large. 

Survivals of T4 virions in selected sole alkaline solutions were examined over extended time courses. Virions in 3 mM NaOH, in 1 mM Ca(OH)_2_, and in 0.1 mM Mg(OH)_2_ solutions maintained 6%, 58%, and 98% of infectivity after one week, respectively. 

### 3.2. H^+^: HCl and H_2_SO_4_

The stabilities of T4 in acid solutions, HCl and H_2_SO_4_, were measured at the range between 10^−2^ mM and 10^3^ mM, corresponding to pH 5 and pH 0 (Figure 1). No pfu were observed at suspensions in whole ranges of concentrations of both acids. As outlined below, when the cations are Na^+^, K^+^, Ca^2+^, or Mg^2+^, the anions Cl^−^ and SO_4_^2−^ do not cause inhibition of pfu formation. It becomes clear that the univalent cation, hydrogen ion, H^+^, inactivates the infectious ability of T4 in the whole spectrum of the concentration. Using the heavier fraction of T4 virions, effects of coexistent ions were examined. Coexistence with other neutral ions, e.g., 0.1 M NaCl or 10 mM CaCl_2_, protects virions from the inactivation by H^+^ at the dilute ranges of H^+^, e.g., lower than 10^−4^ N or higher than nominal pH 4 (Figure 1, dashed lines).

### 3.3. Na^+^: NaOH, NaCl, and Na_2_SO_4_

The effects of the sodium ion, Na+, in the quasi-pure ionic solutions of NaOH, NaCl, and Na_2_SO_4_ were tested (Figure 2). In the alkaline solution of NaOH, ca. 90% of viral activity was maintained at ion levels ranging between 0.1 and 1 mM after 15 min. suspension. No or few pfu were observed below 0.1 mM NaOH (pH 10 equivalent) or higher than 10 mM NaOH (pH 12 equivalent). The pfu values abruptly jumped to 0.1 mM NaOH and greater. 

In the neutral ionic solutions of NaCl and Na_2_SO_4_, the optimal ion levels were 100 mM for NaCl and higher than 1 M for Na_2_SO_4_. No pfu were observed below 1 mM Na_2_SO_4_ or below 10 mM NaCl. There was a clear difference between the optimal concentrations of the alkaline solution, 0.1 mM, and the neutral solutions, 100 mM or higher. The cation, Na^+^, was the same in both cases, so, the difference can be attributed to the anions, the optimal concentration of 0.1 mM OH^−^ in alkaline solution and 100 mM or higher Cl^−^ or SO_4_^2−^ in the neutral solutions. In the neutral ionic solutions, NaCl and Na_2_SO_4_, the infectivity of virions with these neutral ions increased with the increase of the ion concentrations higher than 1 mM.

The pfu profiles of virions in mixed solutions consisting of two sodium ionic compounds, NaOH and NaCl, were obtained using the heavier fraction of T4 virions (Figure 2, a dashed line). The concentrations of NaOH were changed, while the concentration of NaCl was constant, 100 mM, which was the concentration maintaining ca. 40% of the viral infectivity in the pure condition. The pfu profiles were not simple summations of pfu of 100 mM NaCl plus pfu of NaOH. When the concentrations of NaOH were higher than 0.1 mM, 100 mM NaCl did not modify the pfu profile of NaOH. When the concentrations of NaOH were lower than 0.1 mM, the pfu values of the mixed solutions were more than the sum of NaOH and NaCl at the concentration of NaOH was 0.01 mM, and the pfu values were equivalent to the pfu at 100 mM NaCl when the concentration of NaOH was 0.001 mM.

### 3.4. K^+^: KOH, KCl, and K_2_SO_4_

The general activities of the potassium ion, K^+^, in the solutions of KOH, KCl, and K_2_SO_4_ were similar to that of Na^+^ (Figure 2 and Figure 3). In the alkaline solution of KOH, ca. 90% of viral activity was maintained as the ion levels ranged between 0.1 and 10 mM after 15 min. suspension. The pfu values were like 0 or 100, or like an on/off switch, when the concentration of KOH was 0.1 mM (pH 10 equivalent). This causes a very wide error range of this point (Figure 3 and Figure 6). No or very few pfu were observed below 0.1 mM KOH or higher than 10 mM KOH (pH 12 equivalent). 

In the neutral ionic solutions, KCl and K_2_SO_4_, the ability to preserve the infectivity of virions increased with the increase of ion levels higher than 1 mM. No pfu were observed below 1 mM of KCl or K_2_SO_4_. According to the solubility, pfu higher than 100 mM was not tested for the K_2_SO_4_ solution. Clear differences of the pfu profiles were observed between the alkaline KOH solution and the neutral KCl and K_2_SO_4_ solutions. Like Na^+^, the cation, K^+^, was common in these cases, so, the difference can be attributed to the anions, OH^−^ in alkaline solution and Cl^−^ or SO_4_^2−^ in the neutral solutions.

### 3.5. Mg^2+^: Mg(OH)_2_, MgCl_2_, and MgSO_4_

In the divalent cation Mg^2+^, the general profiles of pfu spectra along the concentrations were similar to the profiles of the univalent cations, Na^+^ and K^+^ (Figure 2, Figure 3 and Figure 4). However, in the alkaline solution of Mg(OH)_2_, a clear jump in viral activity was observed at 0.05 mM (pH 9.7 equivalent), not 0.1 mM. The highest preservability of the infectivity of T4 virion was observed at 0.15 mM Mg(OH)_2_, and the preservability was nearly 100% (Figure 4). No or very few pfu were observed below 0.05 mM. According to solubility, concentrations higher than 0.15 mM Mg(OH)_2_ solution were not examined. 

In the neutral ionic solutions, the highest value of the preservability of infectivity was lower than 70% (Figure 4). The preservability increased linearly with the increase of the concentrations of MgCl_2_ and MgSO_4_ on the semi-log plots, and the ion level starting the increase was 0.01 mM, which was two orders lower than the cases of univalent cations, Na^+^ and K^+^. No pfu was observed below 0.01 mM in both MgCl_2_ and MgSO_4_. The highest values of preservation, the initial concentrations appearing the preservation, and the slopes of the increases were clearly different between the alkaline solution, Mg(OH)_2_, and the neutral solutions, MgCl_2_ and MgSO_4_ (Figure 4).

### 3.6. Ca^2+^: Ca(OH)_2_, CaCl_2_, and CaSO_4_

The general profiles of pfu spectra along the Ca(OH)_2_ concentrations were similar to the profiles of pfu in NaOH or KOH solutions (Figure 2, Figure 3 and Figure 5). In the alkaline solution of Ca(OH)_2_, 70–100% of viral activity was maintained as the ion levels ranged between 0.1 and 10 mM after 15 min. incubation. At the concentration of Ca(OH)_2_ as 0.1 mM (pH 10 equivalent), the pfu values were like 0 or 100, or like an on/off switch (Figure 5). No or very few pfu were observed below 0.1 mM Ca(OH)_2_ or higher than 10 mM Ca(OH)_2_ (pH 12 equivalent).

In the neutral ionic solutions, the preservability of the infectious abilities increased linearly with the increase of the concentrations of CaCl_2_ and CaSO_4_ on the semi-log plots. According to solubility, a solution of CaSO_4_ higher than 10 mM was not examined. The level of concentration at which the pfu started to increase was 0.01 mM, which was two orders lower than the cases of univalent cations, Na^+^ and K^+^ (Figure 2, Figure 3 and Figure 5). No pfu was observed below 0.01 mM in both CaCl_2_ and CaSO_4_ (Figure 5). The highest values of preservation, the initial concentrations appearing the preservation, and the slopes of the increases were clearly different between the alkaline solution, Ca(OH)_2_, and the neutral solutions, CaCl_2_ and CaSO_4_ (Figure 5). 

The pfu profiles of T4 virions suspended in mixed solutions consisting of two calcium ionic compounds, Ca(OH)_2_ and CaCl_2_, were obtained with the heavier fraction of T4 virions (Figure 5, a dashed line). The concentration of CaCl_2_ was fixed at 100 mM, which maintained ca. 50% of the viral infectivity (Figure 5), while the concentrations of Ca(OH)_2_ were changed. The pfu profiles were not simple summations of pfu at 10 mM CaCl_2_ and pfu at Ca(OH)_2_. When the concentrations of Ca(OH)_2_ were higher than 0.3 mM, coexistence of 10 mM CaCl_2_ did not modify the pfu profile of Ca(OH)_2_. At the concentration of Ca(OH)_2_ as 0.1 mM, the pfu of the mixed solutions was ca. 100, which was equivalent to the switch-on case of Ca(OH)_2_ (Figure 5). When the concentrations of Ca(OH)_2_ were lower than 0.1 mM, the pfu values of the mixed solutions were more than the sum of Ca(OH)_2_ and CaCl_2_ at the concentration of Ca(OH)_2_ was 0.01 mM, and the pfu values were equivalent to the pfu of 10 mM CaCl_2_ when the concentration of Ca(OH)_2_ was 0.001 mM (Figure 5).

### 3.7. OH^−^: NaOH, KOH, Mg(OH)_2_ and Ca(OH)_2_


To show the abrupt increase of infectivity at around 0.1 mN OH^−^, or pH 10, every count of pfu in alkaline solution in Figure 2, Figure 3, Figure 4 and Figure 5 was plotted against the nominal pH values (Figure 6). The sustainability of the infectivity of T4 was investigated in the range of concentration from 0.01 mN to 100 mN. According to the solubilities, the highest concentrations examined were 2 mN and 10 mN for Mg(OH)_2_ and Ca(OH)_2_, respectively. Little or no sustainability could be observed in the concentrations of OH^−^ below ca. 0.1 mN. The sustainability of the infectivity of T4 jumps up at around 0.1 mN OH^−^; 0.1 mN for NaOH, KOH, and Ca(OH)_2_ and 0.05 mN for Mg(OH)_2_, corresponding to pH 10 and pH 9.7, respectively. In the higher range concentrations of alkaline solutions, the sustainability was suppressed, and no sustainability was observed over 10 mN OH^−^, corresponding to pH 12. 

## 4. Discussion

The effects of a quasi-pure solution of a sole ionic compound on the survival of T4 virions were examined. Previous studies of ionic effects on virion activities were mainly focused on the integrated effects of ions on the virus-host adsorption during pre- and on-adsorption processes in multi-ionic media or buffers [2,4,5,7,8,11]. In this study, the pre-adsorption processes were separated from on-adsorption processes, and the effect of a sole ionic compound on the pre-adsorption process was examined. Virions used in the sole ionic compound were plated and adsorbed to host bacteria following the conventional plating method. Accordingly, in this study the conditions of host bacteria and the adsorption were practically identical for whole cases and the numbers of pfu were the reflections of the irreversible alternations of virions produced during the 15 min. immersion in solutions of the sole ionic compound prior to the adsorption processes. Virions in this study were not exposed to a pure solution of an ionic compound, but rather to a quasi-pure solution, i.e., when an aliquot of viral suspension was inoculated, a trace amount of ions included by the inoculants of virion suspensions were introduced to the test solution. However, the inoculated amounts of ions, i.e., 0.0001 T-buffer: 0.01 mM NaCl, 0.2 μM MgSO_4_, and 0.05 μM phosphate buffer, were far below the minimum concentrations of these ions necessary for pfu formation in the conditions tested (Figure 2, Figure 3, Figure 4 and Figure 5). Indeed, if there were no supplementary ions, the virions lost their activity immediately and no pfu was observed in the tested conditions (Table 1). Therefore, even though the solutions tested were quasi-pure, it was practically a pure solution from the viewpoint of viral responses to ionic conditions.

The relatively short time exposure to a solution of one ionic compound defines the survival of the infection and multiplication processes of T4 virions. It has been known that the optimal divalent cation levels range between 10^−2^ and 10 mM [2,6] and that phages such as T2 and T4 have a requirement of NaCl for adsorption at the level of 100 mM which is ca. one order of magnitude higher than divalent cations [4,8]. In this study, the results produced in quasi-pure neutral ionic solutions include results equivalent to the previous studies which were gained by controlling the target ions in multiple ionic solutions. The increasing curves of the preservability of infectivity along with the increase in the concentrations of quasi-pure neutral ionic solutions were linear to the concentrations of ions in univalent cations. However, they were linear in the semi-log plot in divalent cations (Figure 4 and Figure 5) or the preservability increases logarithmically for divalent cations. That is, the slopes of pfu to the concentrations of the divalent cations were in reverse proportion to the concentrations of the ions, i.e., the preservation of the viral infectivity increased rapidly at the threshold concentration of the divalent cation, and afterwards, the slope of the increase rapidly became gentle. The threshold concentration of the divalent cations to preserve infectivity was ca. 10^−5^ M, which is 10^−2^–10^−3^ times lower than the univalent cations [4]. This implies that maintenance of the infectivity, or deprotonation of DNA in the viral head (see below), may require small amounts of divalent cations which may supply the deficient divalent cations to the pre-existing divalent cations in the viral heads.

Phages generally can show infectious abilities in culture media or in buffers the pH ranges of which are adjusted to 5–11 [1,12,13,14,15,16,19,20]. In this study, virions were exposed directly to “pure” acid or “pure” alkali solutions. Without additional coexisting media or buffers, T4 virions showed no ability to form pfu in any acid to neutral conditions (Figure 1). Because virions were inactivated instantly in pure water [2,4,8], it is not clear if the inactivation in the acidic condition was formed by a passive effect derived from having no preservative agent of infectivity or by an active inactivation of acidic ions. The coexistence of neutral ions, e.g., 0.1 M NaCl or 10 mM CaCl_2_, which had abilities to preserve the activities of virions in ca. 50%, protected virions from inactivation by H+ only when the concentration of H^+^ was lower than 10^−4^ N, or higher than nominal pH 4 (Figure 1). Accordingly, hydrogen ions actively inactivated the T4 virions, and without neutral ions, T4 virions had no tolerance to a sole acid to neutral condition to maintain their infectivity. On the contrary, it has been reported that virions are more stable in suitably high enough alkaline concentrations ranging from pH 7 to pH 11 [13,16,19,20]. In this study, after fifteen minutes incubation, 70–90% of viral survival was maintained in 0.1–1 mM OH^−^ solutions that were equivalent to a range between pH 10 and pH 11. This indicates that in quasi-pure alkaline solutions, T4 virions cannot sustain their infectivity below pH 10, i.e., T4 virions need an alkaline condition, higher than pH 10, to sustain their activity (Figure 2, Figure 3, Figure 4, Figure 5 and Figure 6). In addition to short time sustainability, the time-course changes of viral survival in these solutions indicated that the inactivation coefficients, −*k* in the Equation (1), were compatible in 0.1 mM NaOH, 100 mM NaCl, and 10 mM CaCl_2_ and also −*k* of the sole 0.1 mM Ca(OH)_2_ and 0.1 mM Mg(OH)_2_ were equivalent to or smaller than the full strength T-buffer (Table 1). Pure relevant alkaline solutions can preserve the viral activity for a considerably long time. At around pH 10, the bases of DNA, guanine and thymine, start the deprotonation; pKa of deprotonation of guanine deoxyribose-5′-phosphate and thymidine-5′-phosphate are 9.7 and 10, respectively [21]. In an environment with pH higher than 12, the viral DNA becomes denatured [22]. This indicates a condition in which the bases of the nucleic acids are deprotonated, but the DNA is not denatured; this state is a required condition for maintaining the infectivity of phage virions. The addition of neutral ions, e.g., 0.1 M NaCl to NaOH and 10 mM CaCl_2_ to Ca(OH)_2_, expanded the infectivity toward the lower alkaline concentration range. At 0.01 mN of OH^−^, at which the hydroxy ion had no potential to sustain the infectivity, the addition of neutral ions showed equivalent sustainability with the infectivity at 0.1 mN hydroxy ion (Figure 2 and Figure 5). This implies that these neutral ions may maintain the pre-existed deprotonation of the DNA bases in a viral head (see below).

In the acidic buffers, virions can maintain their infectivity over wider acidic ranges [1,13,16,23]. However, in the quasi-pure acidic condition, T4 virions immediately lose their infectivity (Figure 1). Alcohol-based disinfectants appear to have a minimal effect on non-enveloped viruses, while low-pH alcohols exhibit strong virucidal effects against them [23,24]. Virions tend to irreversibly lose their infectivity in the low-pH range. Contrarily, T4 virions maintain their infectivity in quasi-pure alkaline conditions in the pH range between pH 10 and pH 11 (12) (Figure 2, Figure 3, Figure 4, Figure 5 and Figure 6). When neutral salts coexist in the alkaline solutions, the sustainable range of pH expands into the lower alkaline concentrations (Figure 2 and Figure 5). Calcium hydroxide is used as a virucide at pH 13 in farms [25,26]. The results also indicate the loss of infectivity of T4 virions at a pH range higher than 12. However, alkaline conditions, including calcium hydroxide, show protective actions to the viral infectivity at dilute ranges (Figure 2, Figure 3, Figure 4, Figure 5 and Figure 6). If other ions coexist, the range of protection can expand. Around the area where the alkaline virucidal agent is applied at higher concentration, zones of lower concentration surrounding the area always exist, and some types of virions may be able to maintain their infectivity at these lower concentration zones. While care may be required for using alkaline agents in virucidal applications, the results of a bacteriophage, T4, cannot directly apply to the pathogenic virions. Understanding the activities of virions in ionic solutions is essential for viral dynamics [3,4,5,6,7,8,9]. Prior to investigating the behaviors of virions in the solutions of the mixtures of multiple ions, like buffers, effects of individual ions on viral behavior should be elucidated first. The methodology applied here provides a tool to elucidate the basic effects of individual ions, as well as mixed and combined ions, on the infectivity of viruses.

The minimum required concentrations and pfu-ion concentration curves may be two facets of one process, while the mechanism through which pre-adsorption exposure of virions to ionic solutions causes irreversible changes of infectivity in the subsequent multiplication processes of them is unknown. The viral DNA in capsids maintains compactions through ions like Ca^2+^, and the immersion of virions in ionic solution will exchange ions between the heads and the outer solution [27,28]. This conformational condition of DNA in the head can affect the subsequent multiplication processes of the virions. The range of ionic alternation should include proteins [2,6], while in this report, we concentrated on the ion-DNA correlation. We proposed here that the deprotonation of the bases of DNA, guanine and thymine, is essential for the infectivity of T4 virions. A pH higher than ca. pH 10 is the critical alkaline condition for this deprotonation [21], which reduces the hydrogen bond energy between the double helix DNA chains [29]. The viral DNA becomes denatured when the environmental pH is higher than 12 [22]. In the pH range between 10 to 12, the DNA is not yet denatured but is at the state of lower hydrogen bond energy between the DNA chains. This may be the condition where the viral infectivity is preserved in alkaline solutions. However, the viral infectivity is preserved in a neutral condition when the solvent of the viral suspension contains relevant amounts of neutral ions as discussed above and shown in Figure 2, Figure 3, Figure 4 and Figure 5. These neutral ions are not the acceptor of protons of the bases but may protect the state of the pre-existing deprotonation of DNA. 

The solvent for the thermal denaturation of DNA used in Marmur and Doty (1962) [30] was 0.15 M NaCl plus 0.015 M sodium citrate. The 15 mM of sodium citrate preserved DNA in the coil form, not the compact composition [27]. They emphasized that the solvent of DNA containing the univalent cation as a concentration of 0.15–0.2 M Na^+^ ensures that the DNA in the coil composition will not be denatured. This implies that univalent cations may support the infectivity of virions by preventing the denaturation of DNA in the deprotonation condition in the viral head. In addition, Rao and Black (2010) [31] claimed that T4 DNA is packed in the head with ~1000 molecules of imbedded and mobile internal proteins. The majority of them, the IPI*, distributed high density of basic residues on their surfaces that may allow rapid DNA ejection through the portal and tail without unfolding–refolding. If this state of DNA is derived from the deprotonation of the DNA bases and the IPI* is the major agent of the deprotonation in the neutral ambient condition, the neutral ions may support deprotonation by the IPI* because without suitable concentrations of neutral ions virions lose the ability of infection [1,2,4,6,8,9,13,16]. The maintenance of the deprotonation of DNA in the viral head can be another factor which affects the multiplication of the viruses. 

## 5. Conclusions

Viral activities in quasi-pure ionic solutions were revealed. They were not the simple components as in multi-ionic solutions but elucidated a unique factor in maintaining viral activity responding to their environments. Clear differences were identified in the minimum required concentrations to maintain the infectivity of virions: around 10^−1^ mM for hydroxide ions, 10^−2^ mM for divalent cations, and 1 mM for univalent cations; and the pfu–ion concentration curves of viral preservability: an on/off switch type for hydroxide ions, direct correlation for univalent cations, and a logarithmic curve for divalent cations (Figure 2, Figure 3, Figure 4, Figure 5 and Figure 6). The on/off switch type of viral activity change at pH 10 implies that the deprotonation of DNA bases G and T is the necessary condition for viral infectivity. Further studies are required to elucidate the mechanism of these processes.

## Figures and Tables

**Figure 1 viruses-15-01737-f001:**
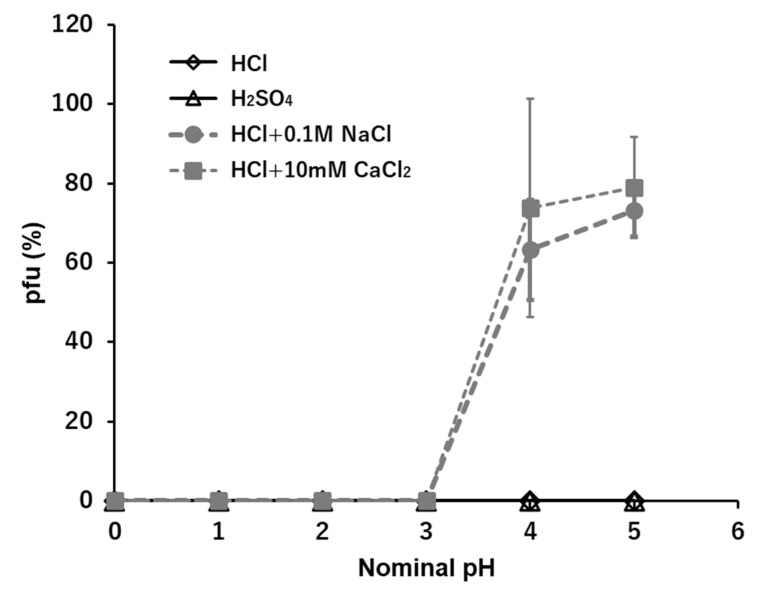
The effects of H^+^, HCl, and H_2_SO_4_ on the infectivity of T4. Dashed lines show the cases of coexistence of neutral ions, 0.1 M NaCl or 10 mM CaCl_2_, in the HCl series.

**Figure 2 viruses-15-01737-f002:**
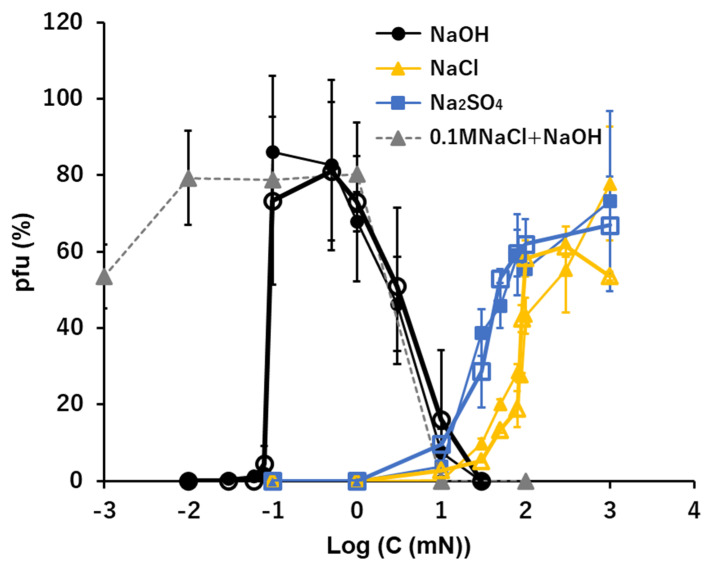
The effects of the solutions of NaOH, NaCl, and Na_2_SO_4_ on the infectivity of T4. Addition of 0.1 M NaCl in the NaOH series is shown with a dashed line. Open symbols: the heavier fraction, Solid symbols: the lighter fraction.

**Figure 3 viruses-15-01737-f003:**
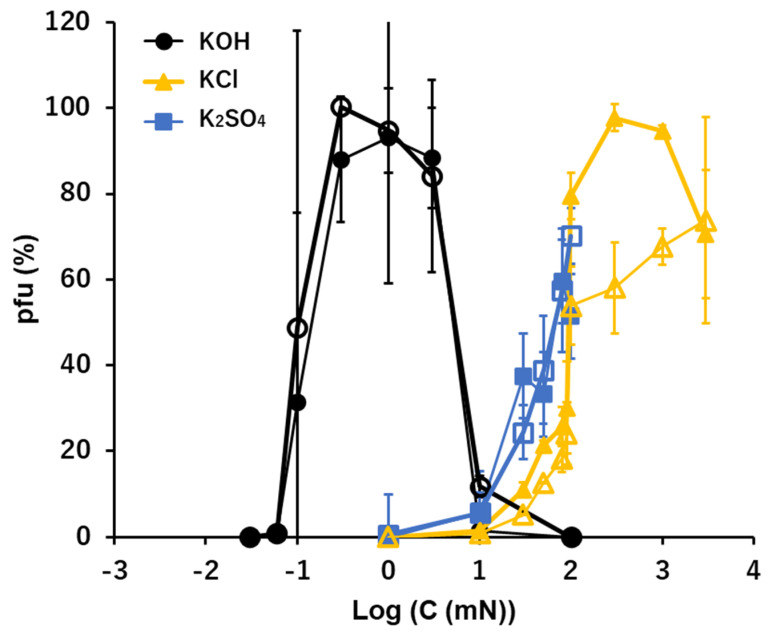
The effects of the solutions of KOH, KCl, and K_2_SO_4_ on the infectivity of T4. Open symbols: the heavier fraction, Solid symbols: the lighter fraction.

**Figure 4 viruses-15-01737-f004:**
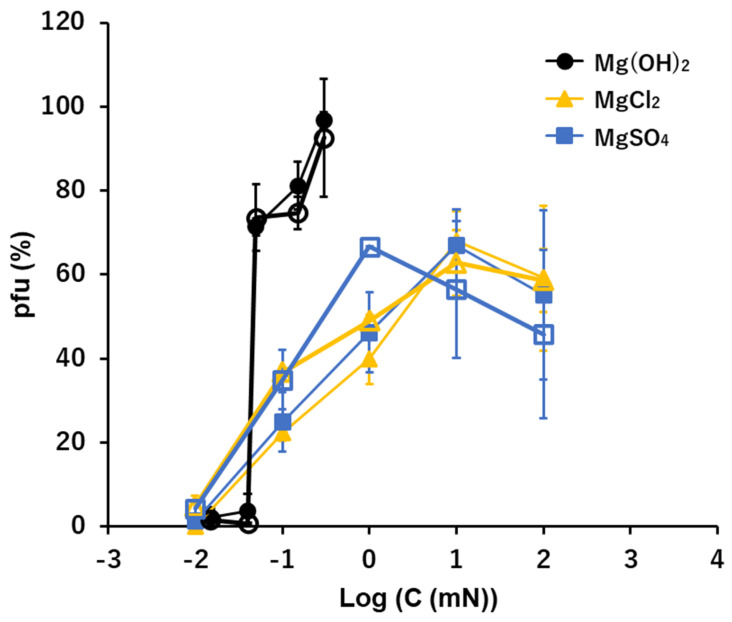
The effects of the solutions of Mg(OH)_2_, MgCl_2_, and MgSO_4_ on the infectivity of T4. Open symbols: the heavier fraction, Solid symbols: the lighter fraction.

**Figure 5 viruses-15-01737-f005:**
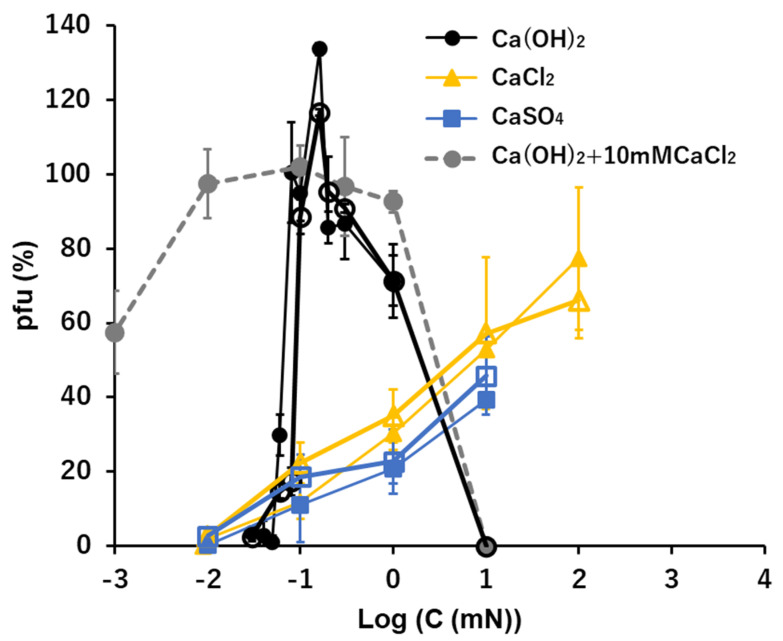
The effects of the solutions of Ca(OH)_2_, CaCl_2_, and CaSO_4_ on the infectivity of T4. Addition of 10 mM CaCl_2_ in the Ca(OH)_2_ series is shown with a dashed line. Open symbols: the heavier fraction, Solid symbols: the lighter fraction.

**Figure 6 viruses-15-01737-f006:**
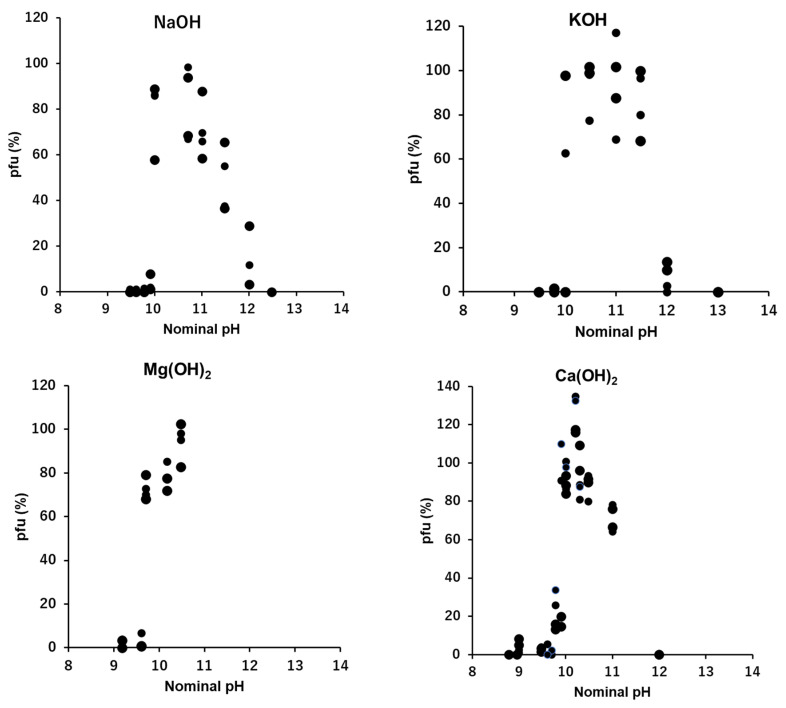
The sustainability of the infectivity of T4 in quasi-pure alkaline solutions. The scales of abscissae are shown in nominal pH.

**Table 1 viruses-15-01737-t001:** The inactivation coefficients in diluted T-buffers and quasi-pure ionic solutions. At 0.0001-fold dilution, virions were inactivated before plating and no pfu were counted. *−k* and *V*_0_ are the inactivation coefficient and the initial abundance of virions in the Equation (1), respectively.

	*−k* (min^−1^) ± S.E.	*V*_0_ ± S.E.	*p* Value of *−k*	R^2^	*n*
0.0001 T-buffer	†	†	†	†	5 (5)
0.001 T-buffer	−3.8 × 10^−1^ ± 2.7 × 10^−3^	311 ± 1.0	4.4 × 10^−3^	1.00	5 (2)
0.01 T-buffer	−1.6 × 10^−2^ ± 5.9 × 10^−4^	78 ± 1.0	1.1 × 10^−4^	1.00	5
1 T-buffer	−4.5 × 10^−5^ ± 3.2 × 10^−6^	89 ± 1.1	3.1 × 10^−5^	0.98	7
100 mM NaCl	−1.4 × 10^−3^ ± 1.0 × 10^−4^	430 ± 1.2	5.2 × 10^−3^	0.99	5
10 mM CaCl_2_	−2.2 × 10^−3^ ± 1.8 × 10^−4^	68 ± 1.1	2.2 × 10^−4^	0.98	6
0.2 mN NaOH	−1.5 × 10^−3^ ± 2.5 × 10^−4^	58 ± 1.2	4.1 × 10^−3^	0.90	7
0.2 mN Ca(OH)_2_	−4.9 × 10^−5^ ± 5.5 × 10^−6^	52 ± 1.1	1.1 × 10^−4^	0.92	8
0.2 mN Mg(OH)_2_	−3.6 × 10^−6^ ± 4.1 × 10^−6^	82 ± 1.1	4.4 × 10^−1^	0.21	6

†: no plaque forming; S.E.: standard error; (*n*): number of specimens with no plaque forming.

## Data Availability

Not applicable.

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
