# Peer review of "Survival of Bacteriophage T4 in Quasi-Pure Ionic Solutions"

_viruses, 2023, doi:10.3390/v15081737_

Round 1

Reviewer 1 Report (Previous Reviewer 1)

Although the manuscript has been reviewed, some points have not been correctly addressed as indicated below:

1. Introduction section.

1.1. From Page 2, lines 45-54: The results obtained in the research (lines 42-43, 46-47, 48-54) must be indicated in the Results section or Abstract section and not in the Introduction. In addition, the objective of the work is not clearly declared in this section. The introduction must include a brief information on the state of the art of the subject investigated, showing the problems encountered and the possible solutions. Finally, and once the subject is focused, the objectives of the work must be declared at the end of the Introduction section. This was indicated in the previous review report.

Instructions on how to write the Introduction section can be read on the journal's website (https://www.mdpi.com/journal/viruses/instructions), as indicated below:

Introduction: The introduction should briefly place the study in a broad context and highlight why it is important. It should define the purpose of the work and its significance, including specific hypotheses being tested. The current state of the research field should be reviewed carefully and key publications cited. Please highlight controversial and diverging hypotheses when necessary. Finally, briefly mention the main aim of the work and highlight the main conclusions. Keep the introduction comprehensible to scientists working outside the topic of the paper.

It is clear that the authors did not follow the rules for writing the introduction section.

2. Materials and Methods section

1.1. Lines 91-95: What statistical software was used to calculate the values of the parameters V0 and k ± standard errors, as well as the statistical significance of each parameter values in equation (1)?

Authors answer: We used the statistic pack of Excel. V0 is the section of the ordinate, and k is the slope of the linear regression. In our study, k has meaning and in the case of k standard errors are not used, but R or R2 are used for similar purpose.

Reviewer answer: The authors used the Excel statistics package. However, with this tool, the standard errors associated to each parameter (V0 and k) could be determined using the linearized form of equation (1):

Ln(Vt) = ln(V0) – k*t

In this case, the standard errors associated with each parameter (V0 and k), as well as the statistical analysis of the parameters and model, are obtained in a separated excel spreadsheet. To do this, click on the “Data analysis” option and when the “Data Analysis” window appears, select the “Regression” option. The Regression summary output is generated in a new excel spreadsheet that includes:

1.     A table with the values of multiple R, R2, adjusted R2, standard error and the number of observations.

2.     An Anova Table with the degree of freedom, sum of squares, mean of squares, F-value and Significance F.

3.     A third table containing the values of the coefficients, their corresponding standard errors, the associated t- and P-values and the Lower 95% and the Upper 95% values for each coefficient.

4.     A fourth Table containing the residual output.

5.     A plot of the fitted regression line and the experimental values.

6.     A graph of the residual output.

On the other hand, the value of R2 is not indicative of the statistical significance of a value of a parameter. In fact, a parameter value is considered statistically significant if its corresponding P-value is lower than 0.05. Surprisingly, some R2 values shown in Table 1 are relatively low (e.g. 0.72, 0.59, 0.26) indicating that the goodness of fit is very poor in these cases. However, the authors did not provide an explanation for this problem. A plot showing both the fitted regression curves and the experimental points should be provided to allow readers to clearly observe the goodness of fit for each curve. Also, an appropriate discussion of these results considering the low values of R2 must be provided.

In any case, linearization of equation (1) to obtain the values of the parameters (V0 and k) is not necessary because the values of the two parameters could easily be obtained using the Solver program in an Excel spreadsheet, although with this procedures, the corresponding standard errors of each parameter are not provided. In this case, it is more convenient to use another statistical program: Statistica, Sigmaplot, etc….

1.2. Lines 94-95: -k is not a parameter, this is an error. The correct form of the parameter is k without the negative sign. The negative sign of this parameter indicated that viral activities decreased exponentially as time increases.

Correct this mistake in the text (lines 94-95) and in Table 1. In this Table, the parameter k is defined as inactivation coefficient or reaction rate coefficient. This should be corrected.

3. Results section.

3.1. Table 1. Page 3, line 132: The values of k should be given as values ± standard errors.

Authors answer: Same as the comment in 1.2.

Reviewer answer: The values ± standard errors of k should be given as indicated in reviewer comment 1.1. With the procedure explained in this comment, the values ± standard errors of k could be easily calculated.

 Based on these comments, I consider that the manuscript in its current form is not acceptable for publication in Viruses.

Author Response

See the attachment

Reviewer 2 Report (Previous Reviewer 2)

The article “Survivals of bacteriophage T4 in quasi-pure ionic solutions” is devoted to fundamental research on preservative qualities of individual ionic compounds on the infectivity of T4.

The article can be printed in Viruses magazine after making minor changes.

I ask the authors to pay attention to the following questions.

In Table 1, the R2 value for 0.1 mM Mg(OH)2 is 0.26. How would you explain this meaning?

Line 42. It is probably not entirely correct to refer to one's own article until it is published. For example, this is done in “[2, 4, 8, this study]”. It might be worth writing that “Similar results were obtained in this study.” It is better to remove the link to this work altogether.

Does the term "our study" refer to this work (mentioned in lines 46-54) or to an already published paper? You need to give a link if the second option. Then this material should be in the “Results” part, if the first option.

The purpose of the work in the “Introduction” part is not specified.

Why are NaCl, Na2SO4, KCl, K2SO4 solutions slightly acidic? These salts are formed by a strong base and a strong acid, so their solutions are neutral.

Why were NaCl, Na2SO4, KCl, K2SO4, CaCl2, CaSO4, MgCl2, MgSO4 chosen for the study?

How correct are the pH measurements in the article using indicator paper, and not a pH meter device? Indeed, for this article, this is a fundamentally important parameter.

How do heavier and lighter fractions differ in chemical and biological composition? Why do they give different results in a biological activity study (eg Figures 2, 3)?

I recommend that the drawings for the heavy and light fractions be placed on different drawings. To make the drawings clearer.

Why is there such a high error for Figure 3 with log(C(nM))=-1.

The number of literature sources with a publication date before 2000 is 52% (15 sources out of 29 cited). Please add additional literature published since 2010.

Round 2

Reviewer 1 Report (Previous Reviewer 1)

Although the manuscript has been revised several times, some points have not been correctly addressed as indicated below:

1. Introduction section.

1.1. From Page 2, lines 27-58: THE OBJECTIVES OF THE WORK HAVE NOT BEEN DECLARED AT THE END OF THE INTRODUCTION SECTION. THIS WAS INDICATED IN THE PREVIOUS REVIEW REPORTS.

Instructions on how to write the Introduction section can be read on the journal's website (https://www.mdpi.com/journal/viruses/instructions), as indicated below:

Introduction: The introduction should briefly place the study in a broad context and highlight why it is important. It should define the purpose of the work and its significance, including specific hypotheses being tested. The current state of the research field should be reviewed carefully and key publications cited. Please highlight controversial and diverging hypotheses when necessary. Finally, briefly mention the main aim of the work and highlight the main conclusions. Keep the introduction comprehensible to scientists working outside the topic of the paper.

It is clear that the authors did not follow the rules for writing the introduction section.

REVIEWER ANSWER 2: THE AUTHORS DID NOT ADDRESS MY EARLIER COMMENT.

2. Materials and Methods section

1.1. Lines 91-95: What statistical software was used to calculate the values of the parameters V0 and k ± standard errors, as well as the statistical significance of each parameter values in equation (1).

REVIEWER ANSWER 2: THE AUTHORS DID NOT ADDRESS MY EARLIER COMMENT.

PLEASE PROVIDE AN APPROPRIATE RESPONSE AND INCLUDE, IN THE MANUSCRIPT, THE STATISTICAL METHOD USED TO CALCULATE THE VALUES OF THE PARAMETERS V0 and k ± standard errors, AS WELL AS THE STATISTICAL SIGNIFICANCE OF EACH PARAMETER VALUES IN EQUATION (1).

1.2. Lines 94-95: -k is not a parameter, this is an error. The correct form of the parameter is k without the negative sign. The negative sign of this parameter indicated that viral activities decreased exponentially as time increases.

Correct this mistake in the text (lines 94-95) and in Table 1 (line 135). In this Table, the parameter k is defined as inactivation coefficient or reaction rate coefficient. This should be corrected.

REVIEWER ANSWER 2: THE AUTHORS DID NOT ADDRESS MY EARLIER COMMENT.

Author Response

Dear Reviewer 1,

Our responses to your review are in the attached file.

Please see the file.

Sincerely youus,

Seiko Hara

This manuscript is a resubmission of an earlier submission. The following is a list of the peer review reports and author responses from that submission.

Round 1

Reviewer 1 Report

The manuscript deals with the evaluation of the survivals of bacteriophage T4 in quasi-pure ionic solutions, to determine the preservative qualities of these ionic compounds on the infectivity of the T4 virions. Although the paper is well written and structured, and the experiments are well planned, some questions need to be explained.

Abstract section.

Page 16, line 12: Define pfu at the first mention.

1. Introduction section.

1.1. From Page 1, line 42 to Page 2, line 56: These two paragraphs should be included in the conclusion section and deleted from the Introduction section. The latter section must be completed with background of previously published papers on this topic, to focus the topic and declare the objectives of the work, which are not stated in this section.  

In short, this section needs to be rewritten, the results of the work removed, and the objectives clearly declared.

2. Materials and Methods section

1.1. Page 3, lines 103-108: Why didn´t the authors measure the pH values of the dilutions using a pH-meter? With this instrument, the pH values of the dilutions are more accurately measured.

1.2. Page 2, line 92: What statistical software was used to calculate the values of V0 and k parameters ± standard errors, as well as the statistical significance of each parameter values in equation (1).

1.3. Page 3, line 110: Why were the plaque forming units (pfu) of each case measured twice and not in triplicate? Usually, triplicate measurements are used in these cases to count the virus.

1.4. Why were the plaque-forming units (pfu) of each case not statistically compared? Why were the experimental data not statistically analyzed?

3. Results section.

3.1. Table 1. Page 3, line 132: The parameter k is not the slope in equation (1). Rather, this parameter is the slope of the linearized form of equation (1):

Ln(Vt) = ln(V0) – k*t

3.2. Table 1. Page 3, line 132: The values of k should be given as values ± standard errors.

5. Conclusions section.

This section is too long. In fact, this section appears to be a discussion section rather than a conclusion section. Commonly in the conclusion section, the references and figures are not mentioned since they are mentioned in the Results and Discussion section. So, in the conclusion section this is not necessary.

It seems that the authors do not know how to write this section. Actually, this section should summarize the main achievements of the work.

Based on these comments, the manuscript in its current form is not acceptable for publication in Viruses.

Reviewer 2 Report

The article “Survivals of bacteriophage T4 in quasi-pure ionic solutions” is devoted to fundamental research on preservative qualities of individual ionic compounds on the infectivity of T4.

The article can be printed in Viruses magazine after making minor changes.

I ask the authors to pay attention to the following questions.

On the signatures of the figures, make superscripts and subscripts in the chemical formulas of the compounds.

Pictures have a great logical meaning in the article. But the use of monochrome black geometric shapes as markers greatly complicates the understanding of the pattern. The lines that show the error when using different reagents are superimposed on each other. I ask the authors to rearrange the drawings.

Figure 6 needs to be checked - some of the OX axes are not printed.

I ask you to arrange the list of cited literature in the same style.

The number of literary sources with a publication date before 2000 is 46% (12 sources out of 26 cited). Please add additional literature that has been published since 2010.

Please indicate the year of manufacture of the instruments used.

In table 1, I ask you to make the Slope -k (min-1) values ​​uniform, for example X * 10-y.

Part of the article "Conclusions" is too large in size. I propose to make it shorter and more capacious in meaning. Probably, part of the text can be moved to "Discussion" .

Why NaCl, Na2SO4, 107 KCl, K2SO4, CaCl2, CaSO4, MgCl2, MgSO4 were chosen for the study?

Very long sentences are used by the authors in the "Discussion" and "Conclusions" parts of the article. Understanding the meaning of the sentence is very difficult.

Part of the article "Conclusions" is too large in size. I propose to make it shorter and more capacious in meaning. Probably, part of the text can be moved to "Discussion" .

Reviewer 3 Report

The article is dedicated to T4 phage virion stability in different solutes, but I find all the experiments to be done in a methodically weak way - for example, no analytical determination of different ions in very dilute solutions was made. More, the authors appear to forget about the proteinaceous part of a virion completely - it's all composed from weak polyelectrolytes able to bind different ions, often in pH-dependent fashion. Neither buffering properties of used solutes were accounted for, nor changing osmotic properties of used test solutions. I strongly recommend to rethink and remake the experiments, so the results would be better interpretable from the viewpoint of protein/DNA chemistry and regular physical chemistry.